# Cytocompatibility and Suitability of Protein-Based Biomaterials as Potential Candidates for Corneal Tissue Engineering

**DOI:** 10.3390/ijms22073648

**Published:** 2021-03-31

**Authors:** Cristina Romo-Valera, Pedro Guerrero, Jon Arluzea, Jaime Etxebarria, Koro de la Caba, Noelia Andollo

**Affiliations:** 1Department of Cell Biology and Histology, School of Medicine and Nursing, University of the Basque Country (UPV/ EHU), Barrio Sarriena S/N, 48940 Leioa, Spain; cristinaromo1@gmail.com (C.R.-V.); jon.arluzea@ehu.eus (J.A.); JAIME.ECHEVARRIAECENARRO@osakidetza.eus (J.E.); 2BIOMAT Research Group, Escuela de Ingeniería de Gipuzkoa, University of the Basque Country (UPV/EHU), Plaza de Europa 1, 20018 Donostia-San Sebastián, Spain; pedromanuel.guerrero@ehu.es (P.G.); koro.delacaba@ehu.es (K.d.l.C.); 3BCMaterials, Basque Center for Materials, Applications and Nanostructures, UPV/EHU Science Park, 48940 Leioa, Spain; 4Department of Ophthalmology, BioCruces Bizkaia Health Research Institute, University Hospital of Cruces, Begiker, Plaza de Cruces S/N, 48903 Barakaldo, Spain

**Keywords:** corneal scaffolds, collagen, gelatin, SPI, cross-linking, cytocompatibility, optical properties, biodegradability

## Abstract

The vision impairments suffered by millions of people worldwide and the shortage of corneal donors show the need of substitutes that mimic native tissue to promote cell growth and subsequent tissue regeneration. The current study focused on the in vitro assessment of protein-based biomaterials that could be a potential source for corneal scaffolds. Collagen, soy protein isolate (SPI), and gelatin films cross-linked with lactose or citric acid were prepared and physicochemical, transmittance, and degradation measurements were carried out. In vitro cytotoxicity, cell adhesion, and migration studies were performed with human corneal epithelial (HCE) cells and 3T3 fibroblasts for the films’ cytocompatibility assessment. Transmittance values met the cornea’s needs, and the degradation profile revealed a progressive biomaterials’ decomposition in enzymatic and hydrolytic assays. Cell viability at 72 h was above 70% when exposed to SPI and gelatin films. Live/dead assays and scanning electron microscopy (SEM) analysis demonstrated the adhesion of both cell types to the films, with a similar arrangement to that observed in controls. Besides, both cell lines were able to proliferate and migrate over the films. Without ruling out any material, the appropriate optical and biological properties shown by lactose-crosslinked gelatin film highlight its potential for corneal bioengineering.

## 1. Introduction

When a large injury or trauma is suffered, the damage caused to the tissues exceeds in some cases their healing capacity, making transplantation necessary to help them restore their function. However, the great imbalance between the demand and available donors and the possible complications related to graft rejection has led to the seeking of alternative solutions. It is in this search for new tissue substitutes where the market and the potential of biomaterials has gotten involved, which is estimated to be close to $150 billion by 2021 [1].

The tissue engineering field is one of the areas in which the applicability of biomaterials could lead to valuable clinical outcomes, where they aim to mimic the structure of native components and extracellular matrix (ECM) to promote cell growth and subsequent tissue regeneration. Protein-based materials are some of the available alternatives.

Collagen is one of the most interesting proteins derived from a tissue source, as it is the most abundant protein in the ECM [2]. In this study, collagen was extracted from porcine skin. By virtue of its various advantages as being biodegradable, biocompatible, highly adaptable, or easily available, the use of collagen and collagen-derived materials has recently been widely increased in tissue engineering applications.

Gelatin is a natural protein derived from partially denatured collagen obtainable from the skin, bone, or connective tissue of all animal species [3]. Its biocompatibility, film formation capacity, and biodegradability are some of the characteristics that make it a suitable alternative for use as implantable biomaterials for regenerative medicine [4,5,6,7]. Likewise, the presence of arginine-glycine-aspartic (RGD) sequences in this polymer facilitate cellular uptake, the functional groups of the amino acid residues allow the structural modification of this material, and its ability to absorb large volumes of water facilitate hydrogels’ formation. Besides, depending on the cross-linking agent used to improve its mechanical behavior, the stability and strength of the resulting structure differ, leading to a material with unique properties. In this work, gelatin was cross-linked with lactose through a non-enzymatic glycation known as Maillard reaction [8] or with citric acid, a tricarboxylic acid that forms amide bonds through the nucleophilic reaction between the amino groups of gelatin and the carboxyl groups of the acid [9].

Plant-derived proteins have also been studied. Soy protein, for example, has shown promising characteristics for its application in the biomedical field. In particular, soy protein isolate (SPI), obtained from soy flour [10], can be physically and chemically modified or combined with other polymers, mainly through its amino and hydroxyl groups [11]. Furthermore, the anti-inflammatory and antioxidant effects offered by its high amount of isoflavones provide properties to promote wound healing [12].

Within this field of tissue engineering, the development of tissue analogues for corneal tissue has become the focus of many current investigations. Naturally-derived materials (e.g., collagen, gelatin, silk fibroin, decellularized corneas, chitosan, hyaluronic acid, etc.), synthetic polymers (e.g., polyethylene glycol-based materials, poly glycolic acid (PGA), acrylate-based polymers, etc.), or composite materials have been proposed for corneal bioengineering approaches with various degree of success [13]. Overall, materials of natural origin tend to be highly biocompatible and can be integrated with no adverse effects; however, they can sometimes increase the chances of rejection by the host tissue or show insufficient mechanical properties. Synthetic materials give the chance to adapt the mechanical and chemical needs to each therapeutic case, but their toxicity evaluation is important because the compounds released when they are degraded can induce adverse reactions that could harm body tissues. Lastly, composite materials aim to combine the advantages of both natural and synthetic materials, such as maintaining biocompatibility while improving strength and controlling degradation rate.

Collagen and its derivatives combined with other materials or crosslinking agents have been the materials of choice of many studies. For instance, Isaacson et al., used sodium alginate and methacrylated type I collagen to prepare bioinks, including corneal keratocytes for developing corneal stromal equivalents. The 3D printed and encapsulated cells demonstrated viability values of 83% seven days post printing [14]. Jumelle et al., used their own patent named GelCORE, consisting of an in situ photo-polymerizable gelatin methacryloyl (GelMA), for healing corneal defects. They incorporated HGF in the developed hydrogel to promote the healing of corneal wounds and tested it in ex vivo models of pig corneal defects [15].

Alternatively, Wang et al. opted for silk fibroin for the development of a corneal model including stroma, epithelium, and innervation. This material allowed them to adjust its mechanical properties and to mold it into transparent films or sponges that could support the growth of neurons and the formation of neuronal connections [16].

The use of composites has also been reported. McTiernan et al. combined collagen-like peptides with polyethylene glycol and mixed with fibrinogen to create an adhesive hydrogel to treat corneal ulcers [17]. Similarly, Rico-Sánchez et al. developed a fibrin–0.1% agarose scaffold containing corneal epithelial cells and human allogeneic stromal keratocytes to mimic the human native anterior cornea. This construct is currently under phase 1/2 clinical trial [18].

Therefore, designing natural or biosynthetic alternatives to human donor corneas has recently gained more attention as it could become a very useful tissue substitute source for overcoming the shortage of supplies, the increasing length of waiting lists and donors’ waiting times, and the rejection limitations related to cadaveric grafts.

One of the strategies to deal with the challenge of mimicking the complex nature of the cornea in terms of mechanical, optical, and biological properties is to focus on the development of scaffolds for the cells of the epithelial, stromal, and endothelial layers. Within this context, the objective of this study was to evaluate the optical adequacy and the cytocompatibility of different natural materials (collagen, soy protein, and gelatin) for their use as corneal scaffolds. To do so, proliferation, adhesion, and migration processes were analyzed in vitro with both epithelial and stromal cells.

## 2. Results

### 2.1. Physicochemical and Morphological Analyses

Fourier transform infrared spectroscopy (FTIR) spectra of all films are shown in Figure 1. The broad band observed in the 3500–3000 cm^−1^ range is attributable to free and bound O–H and N–H groups (amide A), and the absorption peak at 2931 cm^−1^ is attributable to CH_2_ asymmetrical stretching (Figure 1a) [19]. The main absorption bands of proteins are related to C=O stretching at 1635 cm^−1^ (amide I), N–H bending at 1545 cm^−1^ (amide II), and C–N stretching and N–H bending (amide III) at 1241 cm^−1^ (Figure 1b) [20]. Regarding the bands corresponding to the plasticizer, absorption bands of glycerol were located in the region from 800 cm^−1^ up to 1155 cm^−1^, where the absorption band at 1051 cm^−1^ is related to the stretching of C–O bond [20]. Concerning the crosslinkers used in this study, the bands associated with lactose were located between 1180 and 953 cm^−1^ [21], while that related to the carboxylic groups of citric acid appeared at 1743 cm^−1^ [22]. When citric acid reacted with gelatin, the band at 1743 cm^−1^ disappeared, and the gelatin band at 1545 cm^−1^ shifted to 1585 cm^−1^ due to the crosslinking reaction.

In order to investigate the film structure and relate it to the physicochemical properties measured, XRD and SEM analyses were carried out. As can be seen in Figure 1c, collagen (COL) films exhibited a XRD pattern characteristic of partially crystalline materials, with a small peak around 2 θ = 7°, indicating an intermolecular lateral packing distance between the molecular collagen chains and a broad diffuse peak at about 2 θ = 20° due to the diffuse scattering of collagen fibers, representing the amorphous structure of the films. The diffraction patterns of SPI films exhibited a dominant amorphous halo, with a broad band with a maximum at 2 θ = 20°, characteristic of SPI, which has 7S and 11S amorphous globulins as main components [23]. Lactose-cross-linked gelatin (GEL-LAC) films displayed two diffraction peaks: the peak at 21°, related to the crystallinity of gelatin; and the peak at 7.4°, corresponding to the residual triple helix from native collagen. In the same manner, citric acid-cross-linked gelatin (GEL-CA) films exhibited the characteristic peak of gelatin at about 21° and a distinctive diffraction peak corresponding to the triple-helix at a 2 θ value at around 8°. Regarding the film cross-section (Figure 1d–g), all films showed a compact and homogeneous morphology. COL films exhibited a dense fibrillar morphology, in accordance with the lateral packaging of collagen chains observed by XRD analysis. This fibrous morphology became less obvious for gelatin films due to the decrease of the triple helix content.

### 2.2. Light Transmittance and Transparency

COL (Figure 2a) and SPI (Figure 2b) films showed the lowest transmittance values, the dried films being more translucent than the hydrated ones. The differences in light transmission values between the dried and hydrated SPI films decreased as the wavelengths increased, and the differences were no longer statistically significant from 620 nm on. However, the dried collagen film transmitted around 30% more light than the hydrated sample across the entire visible spectrum. In both dried and hydrated SPI films as well as in dried COL films, around 60–80% of light was transmitted in almost all the wavelengths of the visible spectrum. However, the photographs showed blurred patterns, meaning that these films were translucent and allowed the passage of light but did not allow objects placed at a certain distance to be focused clearly (Figure 2e,f,i,j).

Gelatin films transmitted a higher amount of light once hydrated. Light transmission hardly varied in dried or hydrated GEL-LAC films (Figure 2c), and this was the film that transmitted the highest amount of light. The transmission decreased to a low value of 3% in the UVA region. This may indicate a beneficial protective property against this type of radiation. The transmission of light remained low in the wavelengths near the UVA region but increased exponentially as it moved toward yellow (570–580 nm), orange (580–620 nm), or reddish (620–780 nm) regions. The cross-linking procedure used in this film produced a yellowish hue compared with the rest of films, but this did not influence its almost total transparency. The photographed patterns were clear and readable (Figure 2g,k). The greatest light transmission improvement when the film was hydrated was registered for GEL-CA films. Their transparency increased around 20% throughout the visible spectrum, transmitting more than 80% of the incident light from 500 nm on (Figure 2d). This improvement was consistent with the change in transparency since the pattern was clearly recognizable when the film was hydrated (Figure 2h,l).

### 2.3. Degradability

The in vitro degradation profile revealed the behavior of films when exposed to collagenase A enzyme, PBS, or deionized water. Complete degradation of COL films was observed after 15 min immersed in enzymatic solution, probably because it was only physically cross-linked. This complete degradation was translated as very significant statistical differences compared with the control (degradation profile in MilliQ water, Figure 3a). Similar degradation pattern was registered for PBS and MilliQ water solutions, but statistically significant differences were recorded at different time points (*p* < 0.001 at 15 min, 4 h, and 24 h; *p* < 0.01 at 1 h; and *p* < 0.05 at 2 h). Greater weight loss was caused by deionized water than by PBS.

A similar degradation trend was registered for SPI, GEL-LAC, and GEL-CA films (Figure 3b–d). A rapid weight decrease was recorded after 15 min in the presence of collagenase A, PBS, or MilliQ water. The remaining weight of SPI and GEL-LAC films after this time point was maintained around 70% and 80% in the subsequent time steps, and significant differences in relation to control were only recorded after 24 h of enzymatic degradation, when weight percentages decreased up to 40% and 50%, respectively. Regarding GEL-CA films, their weight was reduced up to 60% after being exposed to the three solutions during 15 min, and this weight percentage remained stable until 2 h. The effect of collagenase A became statistically significant in relation to hydrolytic degradation caused by MilliQ water and PBS from 2 h on, when only 30% of the initial weight remained. Unlike COL films, no significant differences between the weight loss caused by the control and PBS solution were observed in the rest of the films. GEL-LAC films suffered the lowest weight decrease in the presence of the enzyme, followed by SPI, GEL-CA, and COL films. The degradation differences between gelatin films highlighted the importance of the cross-linking agent used.

### 2.4. Cell Proliferation and Viability Results

Cell proliferation was studied at 0, 24, 48, and 72 h in 3T3 fibroblasts and human corneal epithelial (HCE) cells seeded in direct contact with the films. Wells with fresh medium and without the biomaterials were used as positive controls. Results showed a time-dependent proliferation pattern in all the wells seeded with 3T3 cells and conditioned with the studied films but with significant differences for wells conditioned with collagen films (Figure 4). Although viability values close to 70% were recorded at 24 h compared with the control, the exposure of the cells to this film in the subsequent time points produced a mild to moderate cytotoxic effect. Statistically very significant differences were registered between the control group and COL films-conditioned cell culture at 48 and 72 h, when an average metabolic activity of 40% and 50%, respectively, were recorded. Viability values recorded for 3T3 cultures in contact with SPI, GEL-LAC, and GEL-CA films were higher than 70% at all the time points. The viability compared with the control at 72 h was 82, 89, and 73% for SPI, GEL-LAC, and GEL-CA films, respectively.

Results concerning HCE cultures also showed a time-dependent proliferation pattern in all cases (Figure 5). Better proliferative pattern and higher viability values were recorded in the HCE-containing wells conditioned with COL films compared with 3T3 cultures. Except for 72 h, when 60% of viability was registered, the viability was much higher than 70% at 24 and 48 h. Even so, the values recorded for COL films were the lowest compared with the other films. The viability recorded for SPI, GEL-LAC, and GEL-CA was well above 70% in all cases.

### 2.5. Cell Adhesion

Cell adhesion and film cyto-compatibility were studied by cell culture observations through the phase contrast microscope, performing the Calcein AM/Ethidium homodimer-1 live/dead assay (EthD1) at different time points (24, 48, and 72 h) and through scanning electron microcopy (SEM) images. The 3T3 cells seeded on SPI, GEL-LAC, and GEL-CA films showed similar cell morphology to the control treatment. They presented the typical bi- and multipolar branched morphology of fibroblasts (Figure 6a–d). Regarding the disposition over the films, a flat and elongated monolayer configuration similar to the control was observed in SPI and GEL-LAC films. However, the cells were arranged in patches in GEL-CA films rather than in a monolayer. Regarding the live/dead Calcein AM-EthD1 assay, very few dead cells were observed above the films; around 1–5% of the cells adhered to the films were stained with EthD-1 (Figure 6e–h). These values were maintained until 72 h. The images taken through SEM showed a very flat and stretched arrangement of these cells over the films. At high magnification, nonsignificant morphological differences were observed among the different samples (Figure 6i–l).

Considering HCE cells, morphological differences were recorded between the cells seeded on different films (Figure 7a–d). The typical cobblestone-like structure of the corneal epithelial cells observed in the control was also recorded in the cell culture on the GEL-LAC film. However, the cells seeded on SPI and GEL-CA films showed a more stretched and rounded cell shape. Similarly, the polygonal outline of cells registered with the control and GEL-LAC films was not so evident. As with 3T3 cells, very few dead cells were observed above the films on live/dead Calcein AM-EthD1 assays performed at 24, 48, and 72 h (Figure 7e–h). The morphological differences of HCE cultures observed by means of phase contrast microscopy were also evident with Calcein AM staining. These cells showed greater cell volume and did not appear as flat as 3T3 fibroblasts through SEM observations. Differences between the control wells and the cells seeded on the different films were neither observed (Figure 7i–l). A stratified arrangement of the cells could be visualized in all conditions.

With respect to both type of cells, cell cultures were clearly observed in GEL-LAC films in the phase contrast microscope. However, it was not possible to focus the cells on the same plane within the same image’s quadrant in SPI and GEL-CA films in some cases due to the films’ surface irregularities. Finally, it was not possible to cultivate any cell type in collagen films; the cells did not adhere to the film so they died and were removed with PBS washes.

### 2.6. Cell Migration

The ability of both cell types to migrate over the films was assessed creating a cell-free gap and recording the time of the gaps’ closure at 0, 24, 48, and 72 h. Technical replicates made with each film type showed the ability of both cell types to migrate over biomaterials. The gaps were closed faster with 3T3 cells than with HCE cells. In both cases, the cells seeded on GEL-CA were the ones that took a longer time for gaps to close; however, all the culture ended up covering the cell-free area by 72 h (Figure 8). As with adhesion assays, it was not possible to cultivate any cell type in COL films.

## 3. Discussion

Numerous materials are currently being assessed as potential candidates to serve as corneal scaffolds or cell support platforms on which cells can adhere and live [13,24,25]. These materials that aim to mimic the native cornea should fulfill the requirements established by the main corneal functions: they should be biocompatible and biodegradable, provide optical properties similar to those of the native tissue, and have the ability to bio-integrate by stimulating the growth, proliferation, and migration of the tissue cells [26]. The current study focused on the in vitro assessment of protein-based biomaterials that could serve as alternative materials for creating corneal scaffolds. This assessment is an essential first step in the design of new materials that could interact with native tissues.

The ability to transmit most of the incident light is one of the essential functions of the cornea. The cornea is responsible for more than 60% of the total refractive power of the eye [27], and this function is strongly linked to its characteristic transparency. In addition to the thinness, morphology, and distribution of the keratocytes in the stroma, the ordered arrangement of collagen fibrils make the cornea a transparent tissue and, in turn, produce destructive interference from scattered light that improves the transmission of light [28]. The artificially generated corneal scaffold substitutes should transmit such an amount of light and resemble the natural structure as closely as possible. That is why the measurement of the optical properties of selected materials is of such importance.

The analysis carried out in this study revealed that the transmittance of the films increased with the increasing wavelength. COL and SPI films transmitted less visible light when hydrated, unlike GEL-LAC and GEL-CA films, in which the hydrated films were the ones that showed the highest amount of light transmission. This indicated that the cross-linking reaction affected the structure and optical properties of the films, as shown by FTIR results.

Concerning COL and SPI films, physical cross-linking by hydrogen bonding between polar groups of proteins and hydroxyl groups of glycerol occurred; however, chemical cross-linking occurred between the amino group of gelatin and the carbonyl group of lactose and the carboxyl group of citric acid. This chemical reaction led to a more compact structure, as shown by SEM images, and to better optical properties.

The total transmittance of light is given by two components: direct transmittance, where the direction of light propagation does not change, and by diffuse transmittance, which is the component that determines the haze and clarity of the objects [29]. To see the effect of diffuse transmission or scattering and to avoid the deceptive contact clarity [30,31], patterns were placed at a distance from the matrices to be tested. It was an observational measure that did not provide quantitative data but was valuable for recording differences between materials. Considering the hydrated state of any material that would be placed on the ocular surface, the transparency acquired by gelatin films was very favorable. By contrast, considering the eye’s essential function of objects’ recognition and focusing, COL and SPI films should be modified to provide better transparency and less blurring since they transmitted between 60% and 80% of the incident light, but they were not translucent enough to recognize patterns clearly. This lack of clarity had to do with the partial structural order of the films that inhibited light scattering [32].

Thus, if transparency would be the major concern, gelatin films would be selected as the best candidates. Gelatin is a versatile material that in itself offers greater transparency than collagen [25]. The gelatin-based gel polymerizable by free radicals in the presence of a type 2 initiator Eosin Y, triethanolamine (TEA), and N-vinylcaprolactam (VC), developed by Shirzaei Sani et al., showed adequate transparency up to 14 days after being applied in rabbit eyes [33]. In the same manner, the gelatin films crosslinked with lactose and citric acid developed in this study demonstrated good clarity and shape definition. Besides, GEL-LAC films specifically could add new beneficial characteristics since the yellowish hue given by the Maillard reaction products, generated after the non-enzymatic glycation of gelatin and lactose [34], provided the benefit of protection against UV radiation. This is a very interesting feature for materials that would be used for the ocular surface. For instance, yellow-tinted lenses have been found to provide a protective function against retinal cell damage [35,36].

Regarding degradation, a suitable material for the cornea should withstand the progressive degradation mechanisms of enzymes in the eye for long enough for the host tissue to regenerate and integrate with the local healthy tissue. The effect of cross-linking was clearly seen since the physical cross-linking occurred in COL and SPI films made them degrade faster than chemically cross-linked gelatin films. In particular, the degradation pattern of collagen decreased exponentially; it was degraded in only 15 min and did not offer resistance to the action of collagenase in vitro. Considering SPI films, soy protein was not degraded by collagenase; it remained stable up to 4 h and then degraded up to 40% after 24 h.

The type of chemical reaction also had influence on gelatin films. After 24 h, around 50% of GEL-LAC remained; in contrast, slightly less than 40% of GEL-CA film was preserved. These results highlight the importance of the cross-linking agent, method, and time to tune the degradation rates of the films. It should not be forgotten that the developed scaffolds or films should provide a suitable substrate and support for cells to grow, migrate, and proliferate during the tissue regeneration process, but these scaffolds should be degraded at the same time that the new corneal stromal ingrowth happens [33,37].

Regarding the cytotoxicity of materials, one of the essential requirements for the biomaterials is the verification of being biologically safe and non-toxic. Within this aim, tests provided by the ISO10993 guidelines for biomedical device evaluation, which are based on the direct or indirect exposure of the material to cell cultures and the comparison of its effects with respect to a control group, can be used [38]. If the effect of the tested material produces a harmful response compared with the limits established with the control, the material is considered unsuitable for use. In this context, the MTT (3-(4,5-dimethylthiazol-2-yl)-2,5-diphenyltetrazolium bromide) tests carried out demonstrated the cellular biocompatibility of SPI and gelatin films with both cell types. On the contrary, the viability values lower than 70% that were registered with collagen films prevented them from being considered a non-harmful material for cells. However, the unsatisfactory results were not attributed to the collagen type I itself. The collagen used in this study was a material that has already been used by other researchers [39,40]. For instance, Isaacson et al., used methacrylated type I collagen mixed with alginate in their constructs and showed viability values of 90% at 24 h and 83% at 7 days with corneal keratocytes, cells that could be equivalent to the fibroblasts used in our study [14]. This underlines the importance of material processing, type of crosslinking, or the addition of components that can completely change the achievable cellular compatibility.

Although statistically significant differences were recorded at 72 h of exposure compared with control wells, it should be taken into account that culture media did not change during this period of time. This condition would not happen in a real environment, where tears and eye blinking would constantly renew the medium exposed to corneal cells [41].

Both cell types of the study were able to be adhered and remained viable over the films. Additionally, the scratch assays revealed that both cell types were able to migrate over the films and that the cell density acquired in the wound area over time was similar to that of the control, especially cells cultured on GEL-LAC and SPI films healed first. Good cell migration and proliferation on gelatin-based materials was also recorded by Shirzaei Sani et al., where it was even higher than in the control [33].

It has been proven that the stiffness of the support material is one of the factors that regulates the behavior and fate of cells. It has been demonstrated that corneal epithelial cells seeded on smooth surfaces more likely retain their undifferentiation potential [42] and that soft surfaces are also interesting for keratocytes to maintain their phenotype since stiffer materials accelerate the transformation to myofibroblasts and, therefore, contribute to an early deterioration of corneal tissue functions [43]. Thus, the differences in cell behavior on the different films suggested the convenience of a study of the surface characteristics of the films, such as roughness, topography, or chemistry, which could influence cell responses in the way they adhere, proliferate, and migrate over the materials [44].

## 4. Materials and Methods

### 4.1. Materials

Porcine collagen was supplied by Tenerias Omega (Navarre, Spain). The treatments of porcine skin to obtain native collagen were carried out according to the method of Andonegi et al. [45]. Soy protein isolate (SPI), PROFAM 974, with 90% protein on a dry basis, was supplied by ADM Protein Specialties Division (Amsterdam, Netherlands). A commercial fish gelatin, with the quality standard for edible gelatin (1999/724/CE), was also employed. Glycerol, used as plasticizer, and lactose and citric acid, used as cross-linking agents, were obtained from Panreac (Barcelona, Spain). Finally, agar was extracted from *Gelidium sesquipedale*.

### 4.2. Preparation of Films

Collagen films were prepared by mixing collagen, glycerol (20 wt %, based on dry collagen), and 0.05 M acetic acid (1:1 collagen/acetic acid ratio). The resulting pastes were stored in a plastic bag for 24 h at room temperature for the dough hydration. The dough was placed between two aluminum plates, put into a Specac press, previously heated up to 90 °C, and then pressed at 0.5 MPa for 1 min to obtain the collagen films, designated as COL.

SPI films were prepared by mixing SPI and glycerol (40 wt % based on dry SPI) for 5 min in order to obtain a good blend. These blends were placed between two aluminum sheets using a caver laboratory press (Specac, Barcelona, Spain), previously heated at 120 °C, and pressed at 4 MPa for 2 min to obtain the SPI films, designated as SPI.

Gelatin films with lactose were prepared by dissolving gelatin and lactose (20 wt % based on dry gelatin) in distilled water for 30 min at 80 °C under continuous stirring to obtain a good blend. After that, 10 wt % glycerol (on gelatin dry basis) was added to the solution, maintained at 80 °C for other 30 min under stirring. Solution pH was adjusted to 10 with NaOH (0.1 M). Finally, solutions were poured into Petri dishes and left drying 48 h at room temperature to obtain films. Additionally, films were heated at 105 °C for 24 h to obtain the lactose-cross-linked gelatin films, designated as GEL-LAC.

Gelatin films with citric acid were prepared by mixing gelatin, citric acid (20 wt % on dry gelatin basis), and agar (10 wt % on dry gelatin basis) in distilled water. Solutions were heated at 80 °C for 30 min and stirred. Then, 20 wt % glycerol (on dry gelatin basis) was added, and solution pH was adjusted to pH 10 with NaOH (0.1 M). The heating procedure was repeated, and finally, solutions were poured into Petri dishes and allowed to cool for 48 h at room temperature to obtain citric acid-cross-linked films, designated as GEL-CA.

Film thickness was measured to the nearest 0.080 mm with a handheld QuantuMike digimatic micrometer (Elgoibar, Spain). Three measurements at different positions were taken from seven specimens for each composition. The calculated average thickness was 0.040–0.050 mm.

### 4.3. Fourier Transform Infrared (FTIR) Spectroscopy

Attenuated total reflectance Fourier transform infrared (ATR-FTIR) spectroscopy was used to identify the characteristic functional groups of the films. Measurements were performed with a Nicolet Nexus FTIR spectrometer equipped with a MKII Golden Gate accessory (Specac) with a diamond crystal as ATR element at a nominal incidence angle of 45° with a ZnSe lens. Measurements were recorded in the 4000–850 cm^−1^ region using 32 scans at a resolution of 4 cm^−1^.

### 4.4. X-ray Diffraction (XRD)

XRD was performed with a diffraction unit PANalytical Xpert PRO (PANalytical, Madrid Spain), operating at 40 kV and 40 mA. The radiation was generated from a Cu-Kα (λ = 1.5418 Å) source, and the diffraction data were collected from 2 θ values from 2 to 90°, where θ was the angle of incidence of the X-ray beam on the sample.

### 4.5. Scanning Electron Microscopy (SEM)

Morphology of the film cross-section was visualized using a Hitachi S-4800 field emission scanning electron microscope (Hitachi High-Technologies Corporation, Madrid, Spain). The samples were mounted on a metal stub with a double-sided adhesive tape. Finally, they were coated under vacuum with gold (JFC-1100) in an argon atmosphere prior to observation. All samples were examined using an accelerating voltage of 15 kV.

SEM visualization of cell cultures was also performed. Samples were fixed using 2% glutaraldehyde in 0.1 M Sorensen buffer with pH 7.4 at 4 °C overnight and then rinsed in Sorensen 0.1 M for three times during 10 min. Samples were dehydrated in increasing concentrations of 30, 50, 70, 90, and 96% ethanol for 30 min each. Two washes of absolute ethanol during 30 min were applied as the last ethanol dehydration step. Dehydrated samples were then immersed into hexamethyldisilazane (HMDS) for 30 min twice and were left in a desiccator to air-dry at room temperature. Once dried, samples were mounted into SEM sample stubs with a double-sided sticky tape. Finally, they were sputtered with gold in argon atmosphere for SEM visualization.

### 4.6. Dialysis and Sterilization

Films were dialyzed by soaking in deionized MilliQ water during 24 h under constant stirring. This step was performed before film sterilization as a first step to remove possible dirt. Once cleaned, they were introduced in the Biolink cross-linker (Vilber Lourmat BLX-254, Collégien, France) for UV sterilization. Samples were located 14.5 cm from the UV light source of 80 W, placed in the internal chamber’s surface of 858 cm^2^, and irradiated for 20 min by each side. The sterilization process was carried out placing the films in the culture plate to be used for subsequent experiments.

### 4.7. Light Transmittance

Light transmission of each film in the visible spectrum (400–800 nm) was evaluated by placing 3 mm diameter disks of each material in a p96-well plate and measuring its absorption in ELx800 Microplate Reader (BioTek^®^ Instruments, Winooski, VT, USA). Light absorption was measured with dried and hydrated films. Wells were filled with PBS to measure light absorption of hydrated films, and empty wells filled with PBS were used as control. Empty wells were used as baseline reference for absorption values of dried films. Measured absorbance values were transformed to transmittance percentages with the following Equation (1):(1)$$\%T=10^{\left(2-\mathrm{Absorbance}\right)}$$

Apart from light transmittance, the transparency of the films was qualitatively measured by analyzing specific photographic patterns placed at a distance of 12 cm from the films.

### 4.8. Biodegradability Assay

Degradation tests were performed to determine weight loss of each film subjected to hydrolytic and enzymatic degradation. To measure the in vitro enzymatic degradation, initially weighed (W_o_) square fractions of dried samples (6 mm × 6 mm) were immersed in a 200 µg/mL collagenase A solution (Sigma, St. Louis, MO, USA) in PBS for different time points (15 min, 30 min, 1 h, 2 h, 4 h, and 24 h) and incubated at 37 °C. At each time point, samples were removed, lyophilized, and weighed (Wt). The in vitro hydrolytic degradation was equally assessed, but samples were incubated in PBS with no enzyme instead. Films soaked in MilliQ water were used as controls. The percentage of degradation was calculated considering the weight loss over time using the following Equation (2):(2)$$\mathrm{Remaining}\;\mathrm{weight}\;\left(\%\right)=\frac{W_{t}}{W_{0}}\times 100$$

All the measurements were performed in triplicates per time point and sample type. Mean values ± SD of the remaining weight percentages were plotted versus time.

### 4.9. Cell Lines

SV-40 immortalized human corneal epithelial (HCE) cells and NIH/3T3 fibroblasts were used for cytocompatibility evaluation. NIH/3T3 fibroblasts were cultured in Dulbecco’s modified Eagle’s medium (DMEM): Ham’s F12 mix (Lonza, Verviers, Belgium) with 10% (v/v) fetal bovine serum (FBS; Lonza), 2.5 mM L-glutamine (Lonza), and 50 U-mg/mL penicillin–streptomycin (Lonza) at 37 °C and 5% CO_2_. HCE cells were maintained under the same conditions as NIH/3T3 cells with the addition of 0.5% (v/v) DMSO (Sigma), 50 µg/mL insulin (Sigma), 10 ng/mL EGF (Sigma), and 0.1 µg/mL cholera toxin (Gentaur Molecular Products, Brussels, Belgium) to the culture medium.

### 4.10. Cell Proliferation and Viability Assay

Cells were seeded in flat bottom 96-well culture plates to perform MTT (3-(4,5-dimethylthiazol-2-yl)-2,5-diphenyltetrazolium bromide) assay. Once attached, cells were starved for 16 h with a solution of DMEM:F12 and 1% of bovine serum albumin (BSA) (Sigma) to synchronize cell cycles. Subsequently, the starving medium was replaced by the corresponding culture or treatment medium according to each assay. Time 0 was set at this step. Cells were seeded in triplicates, and positive controls and treatment blanks were included. Treatment wells were prepared by incubating film sections with the corresponding cell culture media to expose cells directly to the biomaterial tested. Metabolic activity of cells was assessed at times 0, 24, 48, and 72 h. After the corresponding incubation period, the cultures were washed with PBS 1×, and 100 µL of 0.5 mg/mL MTT reagent (Sigma) dissolved in DMEM:F12 was added per well for 3 h at 37 °C and 5% CO_2_. The MTT-containing medium was carefully removed, and 100 µL of DMSO was added per well to dissolve the formazan crystals. Optical densities were determined at 570 nm on a EL × 800 microplate reader (BioTek^®^ Instruments), and absorbance values in treatment and control wells were compared.

### 4.11. Live/Dead Cytotoxicity Assay

The number of alive/dead cells was determined after 24, 48, and 72 h from cells seeding on the films tested using LIVE/DEAD^®^ Viability/Cytotoxicity Assay Kit (Thermo Fisher Scientific, Waltham, MA, USA). A portion of 2 µM of Calcein-AM and 4 µM of EthD-1 was diluted in PBS 1× to prepare the assay’s solution. Culture media was aspirated and a PBS 1× wash was performed on the cells before incubating them for 40 min with the dyes’ solution at room temperature and dark environment. Enough volume of the test solution was added to cover the cells completely. Following incubation, cells were imaged by the fluorescence microscope (Olympus IX71, Olympus, Tokyo, Japan). The test was performed in triplicates, and films without cells seeded on them were included as blanks.

### 4.12. Migration

The migration of HCE and 3T3 cells was studied using two-well culture-inserts (Ibidi, Gräfelfing, Germany) placed above the films. The silicone inserts consisted of two reservoirs for cell culture separated by a defined cell-free gap. Cells were seeded in each reservoir and were left overnight until they attached and formed a monolayer above the films. The defined cell-free gap left by the removal of the insert was registered as the maximum distance between both cell fronts at time 0. Cell migration was then quantified by phase contrast images taken at 24, 48, and 72 h.

### 4.13. Statistical Analysis

Non-parametric Kruskal–Wallis test followed by Dunn’s post hoc test was used for transparency analysis. Degradation and MTT values were subjected to two-way analysis of variance (ANOVA) in which Bonferroni’s test was used for multiple comparisons. Statistical differences were set at *p* < 0.05 level, and all the measurements were performed in triplicates. GraphPad Prism 5 software (San Diego, CA, USA) was used for all statistical calculations.

## 5. Conclusions

The in vitro biological assessment performed has shown that three of the four types of films under study present promising properties as materials for the development of artificial corneal implants, corneal scaffolds, or cell support films. The unsatisfactory results obtained with collagen films were attributed more to the pretreatment of collagen fibers than to the material itself. Although showing adequate biological properties, SPI films would not be the most suitable alternative for implantation in the cornea due to the lack of optical brightness and pattern definition. However, they could serve for the development of scaffolds for the peripheral cornea or other tissues of the body, or even for the central cornea if they were modified to obtain adequate transparency. In sum, gelatin films were found to be the most suitable films to serve as scaffold for corneal tissue substitutes, especially those cross-linked with lactose since they have shown good optical and biological properties. Although this is a first in vitro evaluation, these promising results encourage further study of their potential for corneal tissue engineering.

## Figures and Tables

**Figure 1 ijms-22-03648-f001:**
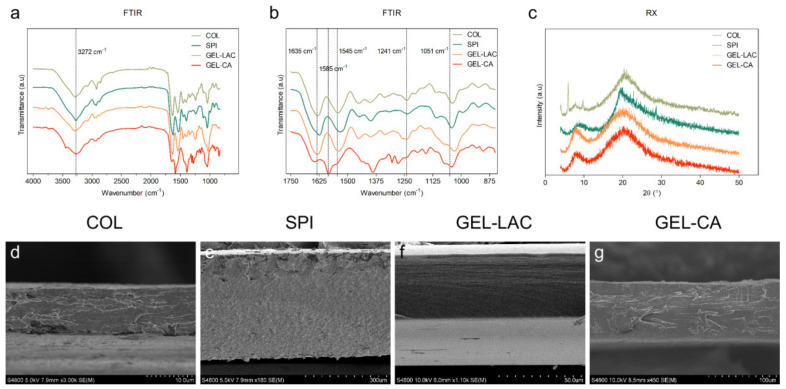
FTIR spectra of films (**a**) from 4000 to 850 cm^−1^ and (**b**) from 1750 to 850 cm^−1^. (**c**) XRD patterns of films. (**d**–**g**) SEM image sections of films obtained with (**d**) 3000×, (**e**) 180×, (**f**) 1100×, and (**g**) 450× magnification. COL, collagen; SPI, soy protein isolate; GEL-LAC, lactose-cross-linked gelatin; GEL-CA, citric acid-cross-linked gelatin.

**Figure 2 ijms-22-03648-f002:**
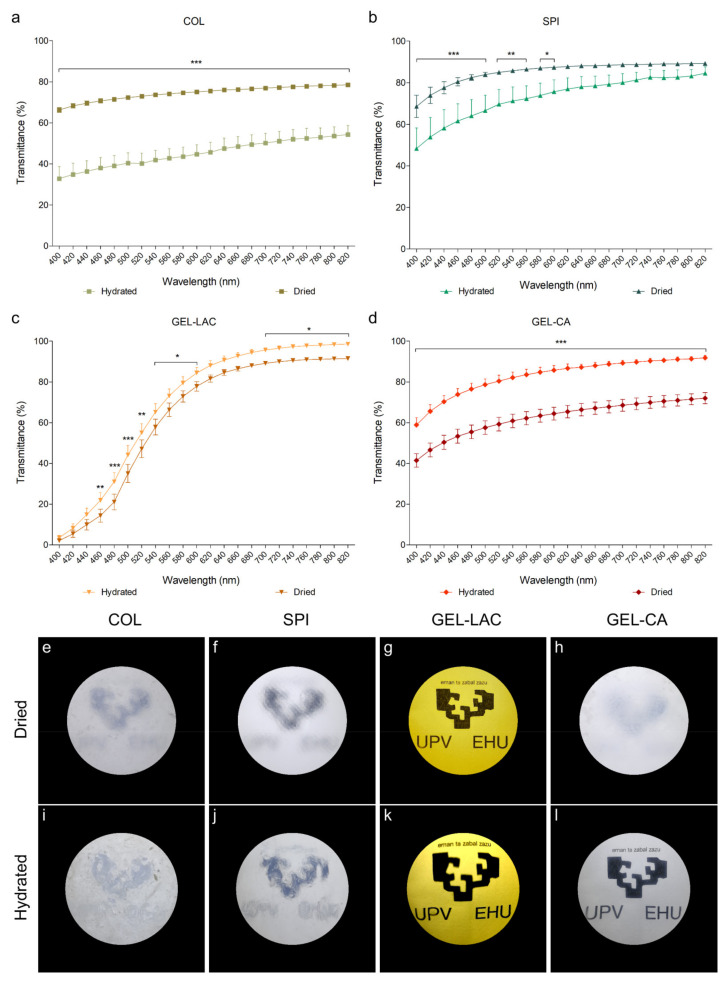
Light transmittance and transparency differences of dried and hydrated films. (**a**–**d**) Graphs show the percentage of light transmitted by each film in the visible spectrum. Data are represented as transmittance percentage mean ± SD. Statistically significant differences show differences of dried with respect to hydrated conditions of each film (* *p* < 0.05, ** *p* < 0.01, *** *p* < 0.001; *n* = 9). (**e**–**l**) Figures show the transparency degree of each film placed at 12 cm from a specific pattern for dried (**e**–**h**) and hydrated (**i**–**l**) films. All the images were taken of the same pattern and with the same conditions.

**Figure 3 ijms-22-03648-f003:**
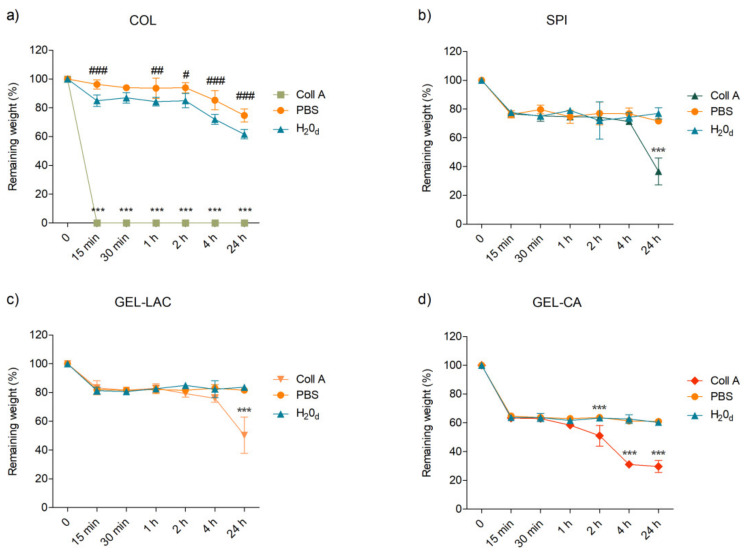
In vitro degradation of the films immersed in 200 ug/mL collagenase A solution or PBS alone at 37 °C over time. Films immersed in distilled water at 37 °C were used as control. **(a)** represents the degradation profiles of COL films; **(b)** the degradation profiles of SPI films; **(c)** the degradation profiles of GEL-LAC films and **(d)** the degradation profiles of GEL-CA films. Data are reported as means ± SD, and statistically significant differences of films in collagenase A (*) and PBS (#) are reported with respect to those in distilled water. * *p* < 0.05, ** *p* < 0.01, *** *p* < 0.001; # *p* < 0.05, ## *p* < 0.01, ### *p* < 0.001; *n* = 3.

**Figure 4 ijms-22-03648-f004:**
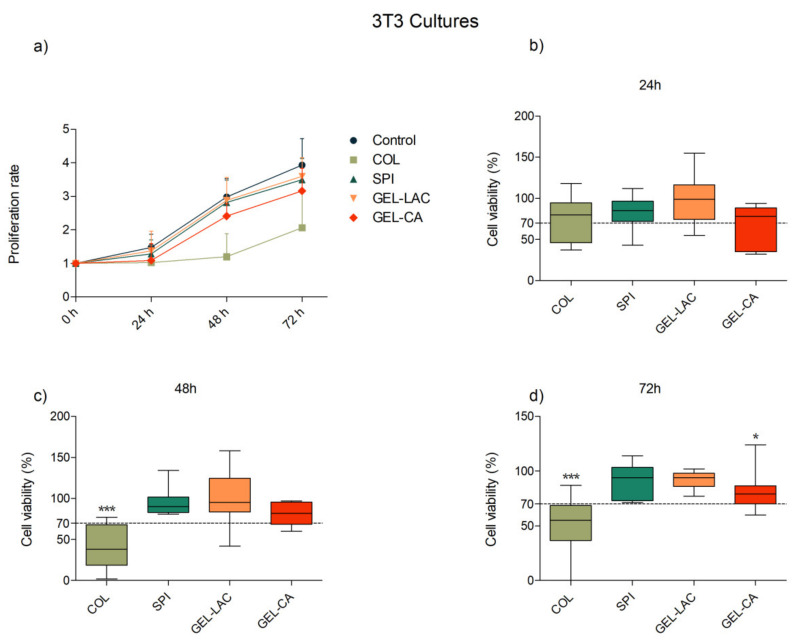
Effects of film composition on the (**a**) proliferation and (**b**–**d**) viability of 3T3 cells. Cells seeded in p96-well plates were exposed to each film for 24, 48, and 72 h. Cells seeded on wells without any biomaterial were used as controls. Proliferation results are represented as proliferation rate mean ± SD of viable cells compared with viable cells at t = 0 h. Viability results are expressed as percentages of viable cells in relation to control wells (100% viability). The lines drawn inside each box represent the medians of viability values of each film. Statistically significant differences showed differences with respect to control (* *p* < 0.05, ** *p* < 0.01, *** *p* < 0.001; *n* = 9).

**Figure 5 ijms-22-03648-f005:**
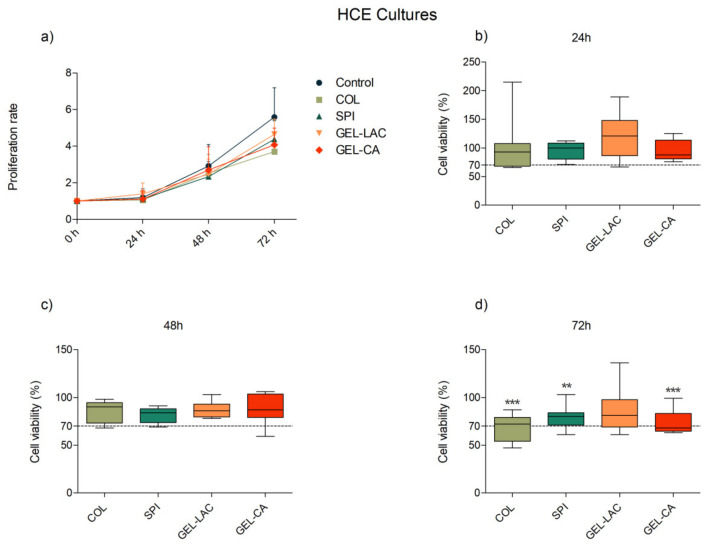
Effects of film composition on the (**a**) proliferation and (**b**–**d**) viability of human corneal epithelial (HCE) cells. Cells seeded in p96-well plates were exposed to each film for 24, 48, and 72 h. Cells seeded on wells without any biomaterial were used as controls. Proliferation results are represented as proliferation rate mean ± SD of viable cells compared with viable cells at t = 0 h. Viability results are expressed as percentages of viable cells in relation to control wells (100% viability). The lines drawn inside each box represent the medians of viability values of each film. Statistically significant differences showed differences with respect to control (* *p* < 0.05, ** *p* < 0.01, *** *p* < 0.001; *n* = 9).

**Figure 6 ijms-22-03648-f006:**
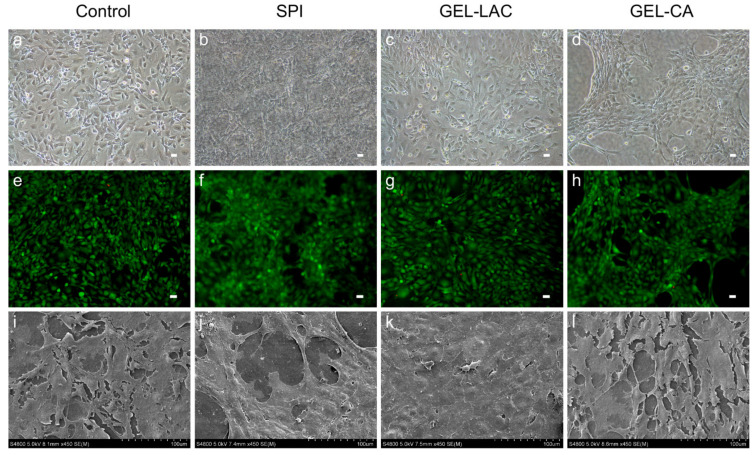
Culture of 3T3 cells above the films. (**a**–**d**) Contrast phase microscopic images of 3T3 cells on tissue culture well plate (control) and the films after 24 h of cell seeding; (**e**–**h**) representative Calcein AM/Ethidium homodimer-1 live/dead (CA-EthD1) assay images and (**i**–**l**) SEM images. (**a**–**h**) Images taken with 10× magnification; scale bars correspond to 25 µm. (**i**–**l**) Images taken with 450× magnification.

**Figure 7 ijms-22-03648-f007:**
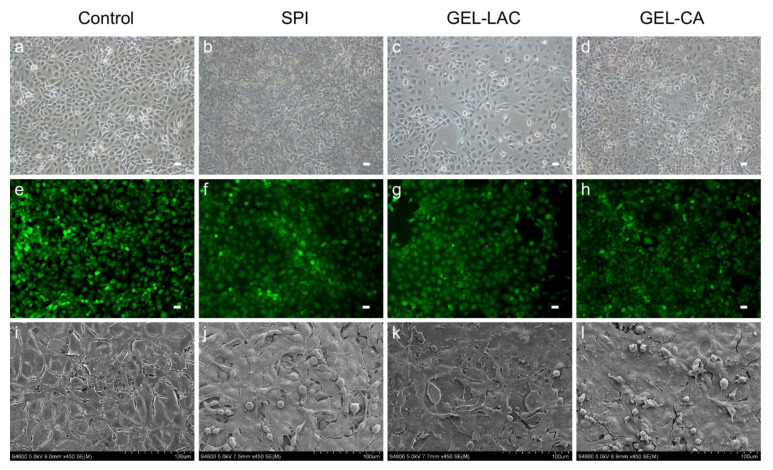
Culture of HCE cells above the films. (**a**–**d**) Contrast phase microscopic images of HCE cells on tissue culture well plate (control) and the films after 24 h of cell seeding; (**e**–**h**) representative CA-EthD1 assay images and (**i**–**l**) SEM images. (**a**–**h**) Images taken with 10× magnification; scale bars correspond to 25 µm. (**i**–**l**) Images taken with 450× magnification.

**Figure 8 ijms-22-03648-f008:**
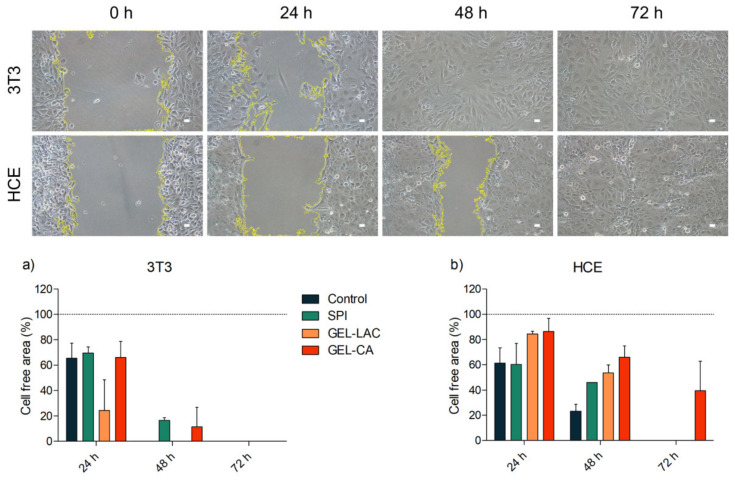
Migration assay of HCE and 3T3 cell seeded above GEL-LAC after 24, 48, and 72 h. Images taken with 10× magnification; scale bars correspond to 25 µm. Data in (**a**) and (**b**) represent the evolution of the gap closure by 3T3 (**a**) and HCE (**b**) cells in each film.

## Data Availability

Not applicable.

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
