# Peer review of "Cytocompatibility and Suitability of Protein-Based Biomaterials as Potential Candidates for Corneal Tissue Engineering"

_ijms, 2021, doi:10.3390/ijms22073648_

Round 1

Reviewer 1 Report

This study analyze four protein-based materials (collagen, soy protein isolate (SPI) and gelatin films cross-linked with lactose or citric acid) as a potential source for corneal scaffolds. First, physical properties of these materials are studied (e. g. FTIR, optical transmittance), and then their degradability, cytotoxicity, cell proliferation and adhesion, were assessed. The manuscript is well structured and written and contains quite new results that can be interesting for tissue engineering. Therefore, it is suitable for publication in IJMS after some revision:

- The scale bars in the SEM images is hardly visible – please improve them (this concerns basically all SEM images presented in the manuscript).

- Designation and description of FTIR spectra is a bit chaotic. In Fig. 1 a and b the bands 1635, 1585, 1545, 1241, 1051 cm-1 are denoted whereas in the text the Authors discuss also the 2931,1155-800, 1180-953, 1743-1690 cm-1 bands, which are not marked in the spectra and do not discuss the origin of the 1585 cm-1 vibration. Moreover, in the region 1743-1690 cm-1, which Authors ascribed to citric acid group vibrations, there are no visible bands (Fig. 1b). Please, improve the analysis. Besides, the peak marking are poorly visible in the FTIR spectra - please use larger  fonts.

- A remark to the interpretation of SEM images: SE detection does not enable to see a “structure” of material. It is to observe solely the morphology of samples.

- What was the thickness of the studied films? Is it important for the in vitro biological assessment (for instance cell adhesion and proliferation)?

Reviewer 2 Report

Manuscript ID: ijms-1143702 

This manuscript, Biocompatibility and suitability of protein-based biomaterials as potential candidates for corneal tissue engineering is an interesting article focusing on the assessment of protein-based biomaterials (collagen, soy protein isolate, gelatin + lactose, gelatin + citric acid) as a potential source for corneal scaffolds. The paper is well-thought although there are several points that may be improved before being accepted in IJMS journal.

[1] In the Title, Abstract, and text, the word biocompatibility should be changed to cytocompatibility. Biocompatibility is a much broader concept and requires more in-depth evaluations.
[2] In the Abstract, you should indicate which of tested biomaterials is the most promising and explain why. The sentence "These results demonstrated that the studied biomaterials could be potential alternative applicable in corneal bioengineering" is too general and does not present the main conclusion of the research.
[3] In the Introduction, please expand the description of the ''current investigations [13-18]". What other biomaterials have been studied? What problems occurred? And why?
[4] In Fig. 1, especially C, improvement of data presentation is required (e.g. normalization, smoothing, and not overlapped mode).
[5] In the caption of Fig. 1, add the magnification for the SEM images.
[6] In the caption of Figs 2, 3, 4, and 5, add the number of repetitions (n=x).
[7] In the Discussion, there is no comparison and discussion of differences/similarities with the results with previously tested biomaterials dedicated also for this or similar applications. For example, form items 13-18?
[8] In the Materials and Methods, add information about the thickness of the films.
[9] In the Materials and Methods, explain why this form of sterilization was used? Why exactly such a UV dose? So far I have experienced other sterilization procedures.

Round 2

Reviewer 2 Report

Manuscript ID: ijms-1143702

This manuscript, Cytocompatibility and suitability of protein-based biomaterials as potential candidates for corneal tissue engineering is an interesting article focusing on the assessment of protein-based biomaterials (collagen, soy protein isolate, gelatin + lactose, gelatin + citric acid) as a potential source for corneal scaffolds. The paper is well-thought, and after the revision presentiation of the results as well as the discussion were improved. The authors responded to all my comments and made the necessary corrections in manuscript. Hence, I recommend accepting the article in present form.